# Graph-Based Deep Learning for Predicting Seizure Outcome in Epilepsy Patients with Thalamic SEEG Contacts

Syeda Abeera Amir[1], Artur A. Aharonyan[1], Marius George Linguraru[1,2],
William D. Gaillard[2,3], Chima O. Oluigbo[2,3], Syed Muhammad Anwar[1,2]

[1]*Sheikh Zayed Institute for Pediatric Surgical Innovation, Children's National Hospital, Washington DC, 20010 USA*
[2]*School of Medicine and Health Sciences, George Washington University, Washington, DC, 20052 USA*
[3]*Center for Neuroscience Research, Children's National Hospital Washington, DC, 20010 USA*

sanwar@childrensnational.org

*Abstract*—**Predicting seizure outcome is essential for tailoring epilepsy treatment. However, accurate prediction remains challenging with traditional methods, particularly in diverse patient populations. This study presents a graph-based deep learning framework for predicting seizure outcomes using stereo-electroencephalography (sEEG) data in pediatric patients with drug-resistant epilepsy and deep thalamic involvement. We analyzed 105 ictal events from sEEG recordings of 10 pediatric patients with documented thalamic seizure networks and evaluated our model in three different cross-validation strategies: seizure-wise, windowed segmentation, and patient-wise analysis. Our graph neural network-based model represents each sEEG channel as a node with power spectral density features, while edges capture inter-channel correlations. The windowed segmentation approach, which divides seizure recordings into non-overlapping 2-second temporal windows, demonstrated superior performance across all metrics. This data augmentation technique achieved 93.9% accuracy, significantly outperforming both seizure-wise (82.%) and patient-wise (77.0%) approaches using complete seizure recordings. Network analysis revealed distinct thalamo-cortical connectivity patterns with denser network topology in sample patient with poor outcomes (<50% seizure reduction) as compared to sample patient with favorable outcomes (>50% seizure reduction). These findings demonstrate the potential of connectivity-based deep learning models for enhancing seizure outcome prediction in pediatric epilepsy, particularly in cases involving complex thalamo-cortical networks. This framework advances our understanding of thalamic seizure propagation and offers promise for AI-assisted personalized epilepsy treatment planning.**

*Index Terms*—**graph learning, network analysis, sEEG, epilepsy, seizure outcome**

## I. INTRODUCTION

Epilepsy affects approximately 50 million people worldwide, with nearly one-third experiencing drug-resistant epilepsy (DRE), where seizures persist despite multiple anti-seizure medications [1]. For these patients, surgical intervention is often necessary to achieve full or partial seizure freedom. Surgical resections can be extensive; however, they are based on the network of brain regions that propagate the seizure. There is strong evidence that surgery can be highly effective in achieving seizure freedom and improving the quality of life in patients with drug-resistant epilepsy [2], [3]. Furthermore, the World Health Organization [1] estimates that up to 70% of individuals with epilepsy could live seizure-free if properly diagnosed and treated, including patients with DRE.

The success of epilepsy surgery heavily depends on the accurate identification of the epileptogenic network and prediction of post-surgical outcomes [4]. This critical need for precise seizure outcome prediction has driven research into developing more sophisticated analytical approaches [4], [5]. Stereo electroencephalography (sEEG) and subdural electrode (SDE) implantation are among the most widely used invasive monitoring methods for identifying the epileptogenic network, with significant evidence supporting better seizure outcomes when utilizing sEEG [2], [6]. sEEG offers deep brain recording capabilities with high temporal resolution and is often used for the identification of discrete cortical seizure onset zones (SOZs) in surgical planning. Among the subcortical regions involved in propagation of seizures, thalamic nuclei have repeatedly shown interconnectedness with ictal brain regions. In a study investigating energy distribution between temporal cortices across seizure stages, the average thalamic power was found to be significantly higher at seizure onset compared to baseline power [2].

The increasing inclusion of thalamic recordings in sEEG implementations has opened new avenues for understanding thalamo-cortical networks, which are fundamental to both developing brain function and pathological states [7]. Despite this wealth of data, traditional analysis methods often struggle to capture the complex, interconnected nature of epileptic networks, particularly the subtle patterns that may predict treatment outcomes. Graph Neural Networks (GNNs) present a promising approach for analyzing such complex neural data, as they can model the brain's networked structure and capture both local and global connectivity patterns [8–11] . In graph-based applications, brain regions are represented as nodes and the strength of their connection as edges, hence enabling the representation of connectivity patterns. Therefore, unlike

traditional machine learning approaches, GNNs can explicitly incorporate spatial relationships and non-linear interactions between brain regions, making them particularly well-suited for analyzing thalamo-cortical connectivity patterns in epilepsy.

In this paper, we present a GNN-based classifier model for predicting seizure outcome using sEEG data. Our contributions include:

1) Our proposed method combines spatial mapping from GNNs with data offering high temporal precision to capture the complex interactions between various brain regions, especially the critical connections between the thalamus and cortical structures.

2) We analyzed the classification of post-resection seizure outcome per patient as well as per individual seizure. We also applied network analysis to study connectivity patterns between the thalamic and cortical regions involved during seizure onset.

3) This approach not only improves prediction accuracy but also provides valuable insights into poorly understood seizure dynamics, such as the relationship between seizure onset zones and seizure networks.

## II. PROPOSED METHODOLOGY

### A. Patient Selection and Implantation

This study investigated sEEG recordings from 10 pediatric patients with drug-resistant epilepsy who had documented thalamic involvement in their seizure networks. The data were collected at our clinical center under an approved IRB study for analyses. All patients underwent stereotactic implantation of depth electrodes with multiple contacts strategically placed within thalamic nuclei, guided by individual clinical presentations and suspected seizure propagation patterns. Figure 1 demonstrates the electrode placement methodology: (a) shows a representative patient's MRI with overlaid electrode trajectories and thalamic contacts, while (b) displays post-operative CT imaging confirming precise contact positioning within deep brain structures.

Seizure outcomes were assessed using a five-point scale [12]: (1) seizure-free, complete elimination of seizures; (2) excellent, $> 80\%$ reduction in seizure frequency; (3) good, $> 50\%$ reduction in seizure frequency; (4) poor, $< 50\%$ reduction in seizure frequency; and (5) worse, worsening of seizures and/or unacceptable neurologic deficit. Based on the post-surgical seizure frequency reduction, patients were stratified into two distinct outcome groups. Group I (Favorable Outcome, n=3): Patients achieving $> 50\%$ seizure frequency reduction, representing successful surgical intervention with significant thalamic network disruption. Group II (Poor Outcome, n=7): Patients with $< 50\%$ seizure frequency reduction, indicating persistent thalamic seizure networks despite surgical intervention.

The dataset comprised 105 ictal events captured across all patients (9-11 seizures per patient), providing substantial data for analyzing thalamic seizure propagation patterns and their relationship to surgical outcomes. Comprehensive clinical information was retrospectively extracted from medical records, including patient demographics, seizure semiology, underlying epilepsy etiology, preoperative diagnostic testing results, non-invasive epilepsy monitoring findings, detailed sEEG procedural records, subsequent surgical interventions, and long-term seizure outcome assessments.

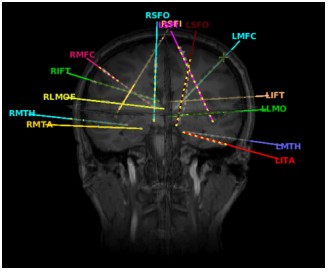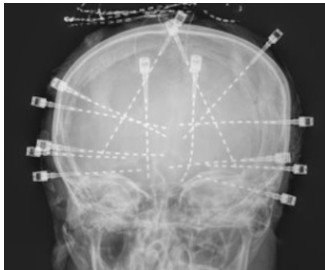

Fig. 1: Demonstration of (a) sEEG electrode placement within the brain and (b) post-op Computed Tomography scan showing sEEG contacts in the brain.

### B. Model Development and Testing

The preprocessing of raw sEEG data involved several steps utilizing MNE-Python [13]. Stereo-EEG data was resampled to a fixed sampling rate of 128 Hz and filtered to include signals between 0.5 and 40 Hz. Each sEEG sample was normalized by subtracting the mean and dividing by the standard deviation of the signal.

Figure 2 shows the model architecture for our pipeline. The processed sEEG data were used to create graph data structures. Each channel was treated as a node, with node features extracted using Power Spectral Density (PSD), and edges were established between all pairs of nodes based on the connectivity matrix of the channels. Connectivity matrix was calculated using Pearson's correlation and thresholded at $\tau = 0.3$, which is a commonly used threshold value to retain nontrivial connections and disregard the rest, i.e., set them to zero [14]. The data was split into training and test sets with stratified sampling to ensure both classes were adequately represented in each set.

The GNN architecture consisted of two graph convolutional layers followed by a global mean pooling layer to aggregate node features. Rectified linear unit (ReLU) activation functions were applied after each convolutional layer. The model's output was either 0, indicating poor seizure outcome, or 1, indicating favorable seizure outcome. We used the cross-entropy loss function and the Adam optimizer for training the model. Hyperparameter tuning was performed using Optuna [15]. The study was run for 50 epochs, with learning rate, number of convolutional layers, hidden dimensions, and dropout rate as the parameters being fine-tuned. The model was trained with K-Fold cross validation (K=10) and evaluated by accuracy, precision, recall, and the F1 score.

### C. Network Analysis

To investigate the spatio-temporal dynamics of thalamo-cortical neural synchronization during seizure onset, we im-

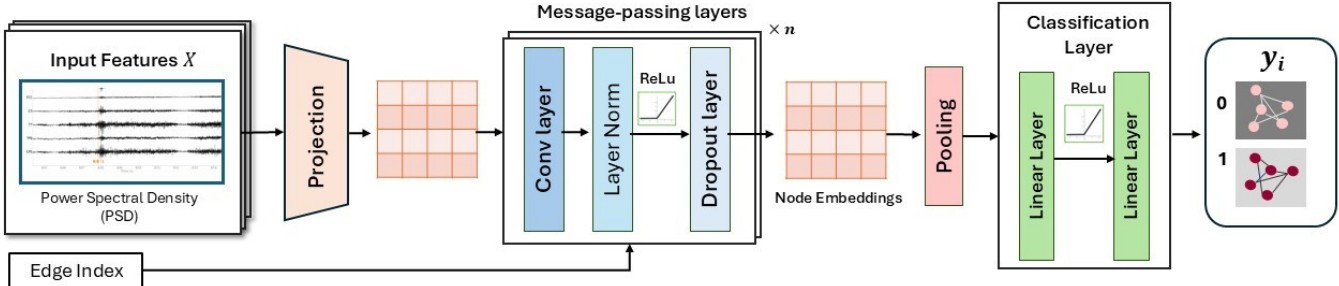

Fig. 2: Our proposed model architecture for seizure outcome prediction. The diagram describes steps for analyzing raw sEEG input, by extracting power spectral density features with a seizure outcome prediction for each segmented graph $i$ with two classes 0: poor outcome, 1: positive outcome.

plemented a network analysis framework. This approach enables systematic visualization and quantification of functional connectivity patterns between thalamic and cortical electrodes, with particular emphasis on characterizing seizure propagation from the seizure onset zone through thalamo-cortical circuits.

Functional connectivity between electrode pairs was quantified using magnitude-squared coherence, a frequency-domain measure that assesses the linear correlation between two signals as a function of frequency. Coherence provides a robust metric for identifying synchronized neural activity while being less sensitive to amplitude variations compared to simple correlation measures. The coherence between two signals $x(t)$ and $y(t)$ was calculated using equation 1

$$C_{xy}(f) = \frac{|P_{xy}(f)|^2}{P_{xx}(f) \cdot P_{yy}(f),} \qquad (1)$$

where $P_{xy}(f)$ represents the cross-power spectral density between signals x and y at frequency f, while $P_{xx}(f)$ and $P_{yy}(f)$ denote the auto-power spectral densities of signals x and y, respectively. The coherence values range from 0 (no linear relationship) to 1 (perfect linear relationship) at each frequency. For network construction, we computed the average coherence across seizure-relevant frequency bands (typically 0.5-40 Hz) to obtain a single connectivity strength value for each electrode pair.

For the network analysis visualizations, we focused on a 5-7 seconds seizure onset range, dividing it into 3 non-overlapping temporal windows ($w = 2s$) to balance temporal precision with statistical reliability of coherence estimates. For each time window, we constructed weighted functional networks where nodes represent individual thalamic and cortical electrode contacts and edges represent coherence values between electrodes. Only connections exceeding a threshold $\theta = 0.25$ were retained to preserve clinically relevant connectivity patterns while reducing noise.

For each time window graph, we computed topological network metrics to characterize evolving connectivity patterns and compare network behavior between patient outcome groups. These metrics include network density, average clustering coefficient, average thalamic node connectivity, and average thalamic node coherence.

## III. EXPERIMENTAL RESULTS

We used three different validation strategies to test the performance of our dataset. For a patient-wise approach we used a Leave-one-patient-out cross-validation (LOPO-CV) where all seizures from a single patient formed the test set while seizures from the remaining 9 patients comprised the training set. In the seizure-wise approach, each seizure was treated as an independent sample, with standard k-fold cross-validation applied across all 105 seizures regardless of patient origin.

In addition to these, we used windowed segmentation with non-overlapping temporal windows applied to segment each seizure into multiple samples, effectively augmenting the dataset size. A total of 2452 graphs were created from the 105 seizures through data segmentation. We used a window size of 10 seconds for the segmentation, with full sEEG recordings ranging from 200 seconds to 600 seconds for all patients. Figure 3 visualizes the data segmentation process.

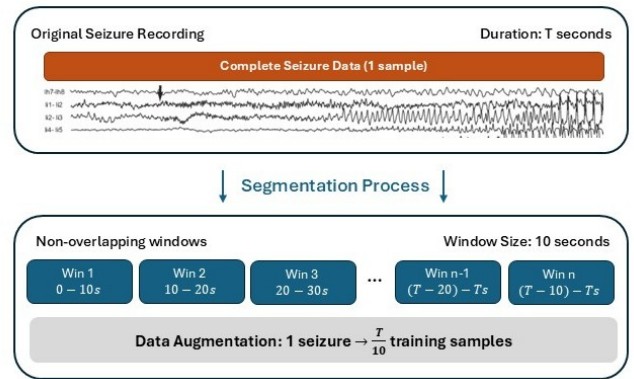

Fig. 3: The segmentation process to create data windows from sEEG recordings.

The experimental results demonstrate significant performance differences across the three analytical approaches for

seizure outcome prediction, summarized in I. The windowed segmentation approach achieved the highest performance across all metrics, with an accuracy of 93.9% Seizure-wise validation showed moderate performance with 82.7% accuracy, while patient-wise validation yielded the lowest performance at 77.0% accuracy.

TABLE I: Performance Metrics for Seizure Outcome Prediction Across Different Experimental Approaches

| Metric | Seizure-wise | Windowed | Patient-wise |
|---|---|---|---|
| Accuracy | $0.827 \pm 0.131$ | $\mathbf{0.939 \pm 0.012}$ | $0.770 \pm 0.155$ |
| Precision | $0.875 \pm 0.099$ | $\mathbf{0.941 \pm 0.012}$ | $0.838 \pm 0.131$ |
| Recall | $0.827 \pm 0.131$ | $\mathbf{0.939 \pm 0.012}$ | $0.770 \pm 0.155$ |
| F1-Score | $0.818 \pm 0.138$ | $\mathbf{0.939 \pm 0.012}$ | $0.756 \pm 0.164$ |

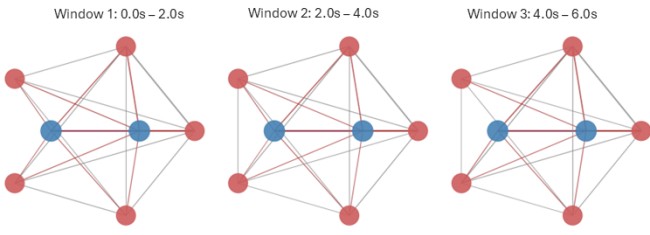

(a) Sample from class 0: poor outcome

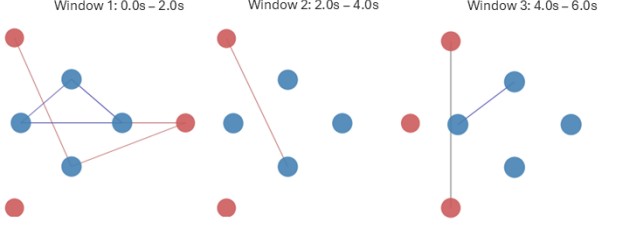

(b) Sample from class 1: positive outcome

Fig. 4: Network dynamics at seizure onset from samples (a) with poor outcome, (b) with positive outcome post-surgery. The figure shows thalamic nodes in blue and cortical seizure onset zone nodes in red for both patients. Edges are calculated using coherence, thresholded at $\theta = 0.25$. Within the region (cortical-cortical and thalamic-thalamic) connections are highlighted in red and blue edges, respectively, and the thickness of the edges represents the strength of the connection. The same subset of nodes is used in all three windows to demonstrate seizure dynamics in one network.

Figures 4a and 4b visualize the evolution of thalamo-cortical network dynamics at seizure onset for two representative patients, one from each outcome group (favorable vs. poor outcome). The two figures highlight significant differences in network topologies over time, quantified in table II for three consecutive time windows (W1, W2, W3). Patients with poor surgical outcomes demonstrated consistently high network density values, ranging from 0.952 to 1.00 across all time windows, indicating dense connectivity between thalamo-cortical regions during seizure propagation. In contrast, patients with favorable outcomes exhibited significantly lower network density, with values of 0.285, 0.047, and 0.095 for windows W1, W2, and W3, respectively.

The role of thalamic nodes was significantly different between the two sample patients as well. Poor outcome patient maintained consistently high thalamic connectivity with an average degree of 5.8 across the three time windows compared to an average degree of 1.6 in the patient with a positive outcome. Average thalamic node coherence values were moderately elevated in poor outcome patients (0.404-0.432) compared to favorable outcome patients (0.00-0.308).

## IV. DISCUSSION

Our model achieved high performance across all three experimental approaches, with the windowed segmentation method demonstrating superior results at 93.9% accuracy, followed by seizure-wise analysis at 82.7% accuracy, and patient-wise analysis at 77.0% accuracy. The windowed approach also showed the most consistent performance with low standard deviation ($\pm 0.012$), indicating robust and reliable predictions across different data segments.

Thalamo-cortical network analysis revealed distinct connectivity patterns between patient outcome groups. Patients with poor surgical outcomes (Class 0) exhibited dense network connectivity with network density values ranging from 0.952 to 1.00 across time windows, accompanied by high thalamic node degrees (5.5-6.0) and elevated clustering coefficients. In contrast, patients with favorable outcomes (Class 1) demonstrated significantly lower network density (0.047-0.285), reduced thalamic connectivity (0.25-2.25 average degree), and minimal clustering, particularly in later time windows. This suggests that excessive thalamo-cortical connectivity during seizure onset may contribute to poorer surgical outcomes, possibly reflecting more widespread seizure propagation that extends beyond surgically treatable regions.

The superior performance of the windowed segmentation approach demonstrates the value of temporal data augmentation in capturing seizure dynamics. By dividing seizure recordings into 2-second non-overlapping windows, the model effectively increased the training dataset while preserving clinically relevant temporal patterns. This finding has important implications for clinical implementation, as it suggests that shorter data segments can provide sufficient information for accurate outcome prediction.

One notable strength of our study is achieving high accuracy with a relatively small cohort of only 10 pediatric patients with deep thalamic contacts. This demonstrates that connectivity-based deep learning approaches such as GNNs may be particularly effective for specialized patient populations where traditional large-scale studies are challenging. The ability to extract meaningful patterns from limited data suggests promise for applications in rare epilepsy syndromes and personalized treatment planning where patient populations are inherently constrained.

TABLE II: Network Properties of Thalamo-cortical Activity Across Time Windows for Coherence Threshold $\theta = 0.25$.

| Metric | Class 0 (Poor Outcome) | | | Class 1 (Favorable Outcome) | | |
|---|---|---|---|---|---|---|
| | W1 | W2 | W3 | W1 | W2 | W3 |
| Density | 0.952 | 1.00 | 0.952 | 0.285 | 0.047 | 0.095 |
| Average Clustering | 0.952 | 1.00 | 0.952 | 0.333 | 0.000 | 0.00 |
| Average Thalamic Node Degree | 6.0 | 6.0 | 5.5 | 2.25 | 0.25 | 0.50 |
| Average Thalamic Node Coherence | 0.408 | 0.404 | 0.432 | 0.308 | 0.308 | 0.00 |

Utilization of sEEG data in conjunction with advanced deep learning techniques, like we present in this study offers a highly promising direction for personalized seizure treatment planning. Enhancing model accuracy through the incorporation of additional modalities such as MRI and PET could also provide a more comprehensive understanding of epilepsy. However, variations in individual patient characteristics, such as the type and location of epilepsy, may not be fully captured in a small sample size such as ours. Future studies that include larger patient populations can refine the model's performance, ensuring its robustness and reliability in different clinical settings. Expanding the sample size would also enable the exploration of more nuanced patterns and relationships within the data, ultimately enhancing the model's predictive capabilities and clinical utility.

## V. CONCLUSIONS

This study presents a novel connectivity-based deep learning framework that demonstrates the potential for accurate seizure outcome prediction in pediatric patients with complex thalamo-cortical epilepsy networks. By integrating graph neural networks with temporal segmentation strategies, we have shown that machine learning approaches can effectively capture the complex dynamics of seizure propagation through deep brain structures, even with limited patient cohorts typical of specialized epilepsy populations. Our findings establish that thalamo-cortical connectivity patterns during seizure onset contain predictive information about surgical outcomes in patients with wide/complex seizure networks or multi-focal epilepsy, providing a foundation for pre-operative assessment tools. Future research with more patients and multi-modal prediction pipeline provides a promising direction in this field.

The ability to achieve robust predictions with a small, specialized patient population demonstrates the particular value of connectivity-based deep learning for rare epilepsy syndromes and complex cases involving deep brain structures. This approach addresses a critical gap in personalized epilepsy care, where traditional large-scale machine learning methods may be impractical due to the inherently limited patient populations. Beyond immediate clinical applications, this work establishes a framework for investigating brain connectivity disruptions during pathological states. The integration of spectral analysis with graph-based network representations offers a powerful methodology that may extend to other neurological conditions involving network dysfunction. As AI-assisted clinical tools continue to evolve, connectivity-based approaches like those presented here represent a promising direction for advancing

personalized neurological care and deepening our understanding of brain network dynamics during disease states.

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
