# OpenReview forum: "Graph-Based Deep Learning for Predicting Seizure Outcome in Epilepsy Patients with Thalamic SEEG Contacts"
_IEEE.org/EMBS/BHI/2025/Conference — BHI 2025_

### Official Review · Reviewer_yXJy · 2025-07-03
**Graph-Based Deep Learning for Predicting Seizure Outcome in Epilepsy Patients with Thalamic SEEG Contacts**

**Confidence:** 4
**Clarity Of Writing:** great
**Clinical Significance:** good
**Methodological Novelty:** great
**Overall Rating:** 7

**Experiments And Results:**

good

**Questions For The Authors:**

Why was a 2-second window chosen for segmentation? Were other time segements investigated?

How was seizure onset and termination determined for each event?

How was the edge threshold selected for graph construction? Was any sensitivity analysis conducted?

**Strengths:**

The approach was able to achieve high predictive performance using data from a small cohort of only 10 pediatric patients with sEEG recordings. This is promising in the context of rare or complex neurological conditions, where collecting large datasets is often not feasible. The paper's approach demonstrates that graph-based deep learning, when paired with domain-specific features and smart data augmentation (temporal segmentation), can extract meaningful patterns from limited data.

The use of GNNs is well-matched to the problem domain, as seizures are fundamentally network-level events. Modeling sEEG contacts as nodes and inter-channel relationships as edges allows the model to capture both local and global aspects of thalamocortical connectivity. This biologically inspired representation is clear improvement over traditional ML methods that treat channels independently.

Beyond classification, the study provides interpretable network metrics that distinguish favorable vs. poor surgical outcomes. These features offer potential biomarkers for surgical candidacy, which can aid clinical decision-making.

**Summary Of The Paper:**

The paper presents a graph-based deep learning framework to predict post-surgical seizure outcomes in pediatric patients with drug-resistant epilepsy involving thalamic networks. The authors use stereo-electroencephalography (sEEG) data from 10 patients with electrodes implanted in cortical and thalamic regions. A graph neural network is trained to classify seizures into favorable or poor outcome groups. The model was evaluated using three cross-validation strategies: patient-wise, seizure-wise, and a windowed segmentation approach that splits seizures into 2-second non-overlapping segments.

The windowed segmentation approach achieved the highest accuracy and functional network analysis using coherence revealed that poor outcome patients exhibited denser thalamocortical connectivity, higher thalamic node degree, and greater network clustering during seizure onset compared to favorable outcome patients. The findings suggest that GNNs, combined with temporal segmentation and coherence-based network analysis, can predict surgical outcomes in complex thalamic epilepsy cases.

**Weaknesses:**

While the paper methodology uses multiple augmentation and validation methods to address the small cohort, this remains a central limitation. Even with data augmentation, all samples are ultimately nested within 10 individuals. As seen in the lower performance of the patient-wise cross-validation, the model likely captures intra-patient patterns better than truly generalizable inter-patient relationships.

The model segments seizures into 2-second windows, treating each as an independent graph. While this improves performance, it might ignore temporal dependencies between windows, particularly the evolution of seizure dynamics over time.

While the paper includes functional network metrics to compare outcome groups, it lacks direct interpretability of model predictions. The paper could, for example, incorporate GNN interpretability tools like GNNExplainer, PGM-Explainer, or saliency mapping to highlight which nodes and edges contributed most to a prediction. This could offer important mechanistic inisights such as identifying specific thalamic contacts or cortical connections that consistently predict poor outcomes and increase the clinical significance of the work.

---

### Official Review · Reviewer_obm3 · 2025-07-07
**Graph-Based Deep Learning for Predicting Seizure Outcome in Epilepsy Patients with Thalamic SEEG Contacts**

**Confidence:** 2
**Clarity Of Writing:** great
**Clinical Significance:** great
**Methodological Novelty:** good
**Overall Rating:** 6

**Experiments And Results:**

good

**Questions For The Authors:**

1. How generalizable do you expect your results to be for adult populations or patients with other forms of epilepsy? This could enhance understanding of the model’s applicability and robustness.

2. Could you discuss potential limitations of the chosen coherence threshold (θ = 0.25) and its impact on your results? Addressing this may influence confidence in your connectivity pattern findings.

3. Could the inclusion of additional data modalities (e.g., MRI, PET) further enhance model accuracy and clinical relevance? Insights into future directions or multimodal integration could substantially improve the methodological rigor.

4. How differences in seizure types (e.g., focal vs. generalized) may affect the predictive power of your model? This would help determine how adaptable the model is across different epilepsy subtypes and whether seizure heterogeneity impacts performance.

**Strengths:**

Clear and clinically important research focus on epilepsy treatment outcome prediction. Innovative integration of GNNs with high-resolution sEEG data to capture complex neural connectivity. Robust methodological design, including multiple cross-validation strategies and rigorous hyperparameter optimization. Strong experimental results with notably high predictive accuracy using windowed segmentation. Comprehensive network analysis providing deeper insights into epilepsy propagation dynamics. Valuable implications for personalized epilepsy treatment and surgical planning.

**Summary Of The Paper:**

The paper presents a novel graph-based deep learning framework utilizing stereo-electroencephalography (sEEG) data to predict seizure outcomes in pediatric patients with drug-resistant epilepsy involving thalamic networks. By representing sEEG channels as nodes and inter-channel correlations as edges in a graph neural network (GNN), the authors effectively model thalamo-cortical connectivity patterns. The model demonstrated strong predictive capabilities, particularly when employing a windowed segmentation data augmentation strategy, achieving an accuracy of 93.9%, significantly outperforming other validation methods. Network analysis also revealed distinct connectivity patterns between patients with favorable versus poor surgical outcomes.

**Weaknesses:**

Small patient cohort (n=10), potentially limiting generalizability. Limited validation at deeper hierarchical or finer temporal scales of neural dynamics. Potential bias introduced by using only pediatric cases with deep thalamic contacts. Lack of external validation with independent datasets or larger clinical populations.

---

### Official Review · Reviewer_JgNn · 2025-07-15
**Graph-Based Deep Learning for Predicting Seizure Outcome in Epilepsy Patients with Thalamic SEEG Contacts**

**Confidence:** 4
**Clarity Of Writing:** great
**Clinical Significance:** great
**Methodological Novelty:** good
**Overall Rating:** 6
**Final Rating:** 6

**Experiments And Results:**

fair

**Questions For The Authors:**

- What is the average number of seizures recorded per patient? This information can help readers assess potential over-representation of certain individuals in seizure-wise approach and windowed-approach
- I assume that the same outcome label is applied across all seizures from a given patient. Can authors confirm this?

**Strengths:**

- The rationale for adopting a graph-based approach is clearly stated in the introduction and is grounded in neurobiological interactions
- I appreciated the detailed description of preprocessing, feature extraction, model architecture, and the hyperparameter tuning process
- The inclusion of a network analysis framework using coherence metrics to investigate thalamo-cortical connectivity was interesting
- Specific network metrics are clearly described
- The use of leave-one-patient-out cross-validation is an appropriate methodological choice given the small dataset size
- The network analysis findings are interesting and may have potential applications in investigating brain connectivity disruptions during pathological states

**Summary Of The Paper:**

The paper introduces a graph-based deep learning framework designed to predict post-surgical seizure outcomes (<50% vs >50% reduction in seizure frequency) in pediatric patients with drug-resistant epilepsy involving the deep thalamus. The model utilizes stereo-electroencephalography data and incorporates a network analysis of thalamo-cortical connectivity at seizure onset.

**Weaknesses:**

- I recommend explicitly reporting the final hyperparameter values in Section II.B. to enhance reproducibility
- I am concerned whether treating individual seizures as independent samples (or using windowed segmentations), irrespective of patient origin, is reasonable. This approach risks patient-level data leakage, especially when evaluating on a per-seizure basis. It may confound model performance by learning subject-specific artifacts rather than generalizable patterns. Perhaps a windowed data augmentation approach combined with patient-wise LOPO-CV can be more appropriate?
-  I suggest including a statistical testing of thalamic connectivity metrics between the outcome groups. Demonstrating significance (or lack thereof) would help validate the utility of these measures

Minor
- The statement “The data was split into training and test sets with stratified sampling…” is unclear without specifying the exact ratio and whether class balance was considered. If the intention was explained in the first paragraph of Section III, consider moving that information to the Methods section for clarity.

---

### Official Review · Reviewer_yJtJ · 2025-07-16
**Graph-Based Deep Learning for Predicting Seizure Outcome in Epilepsy Patients with Thalamic SEEG Contacts**

**Confidence:** 3
**Clarity Of Writing:** good
**Clinical Significance:** great
**Methodological Novelty:** great
**Overall Rating:** 7

**Experiments And Results:**

good

**Questions For The Authors:**

What was the rationale behind selecting the correlation threshold (τ = 0.3) and coherence threshold (θ = 0.25), and have you explored sensitivity analysis to these parameters?

**Strengths:**

Novel Application
Achieves strong classification accuracy (93.9%) despite working with a relatively small and specialized dataset, demonstrating the potential of connectivity-based methods.
Provides a well-structured pipeline, including preprocessing steps, model architecture, hyperparameter tuning, and performance metrics.

**Summary Of The Paper:**

The paper proposes a graph-based deep learning framework using GNNs to predict post-surgical seizure outcomes in pediatric patients with drug-resistant epilepsy involving thalamic networks, based on stereo-EEG (sEEG) recordings. Three validation strategies were compared: patient-wise, seizure-wise and a windowed segmentation approach that divided seizures into 2-second windows, which achieved the highest accuracy compared to seizure-wise and patient-wise methods. Network analysis revealed that patients with poor surgical outcomes exhibited denser thalamo-cortical connectivity during seizure onset compared to those with favorable outcomes. The results suggest that GNN-based models can effectively capture complex brain connectivity patterns and may support personalized epilepsy treatment planning, although the study is limited by its small sample size and lack of external validation.

**Weaknesses:**

The approach does not clarify whether outcome classes were balanced or if measures were taken to address imbalance.